# ORTHOGONALISING GRADIENTS TO SPEEDUP NEURAL NETWORK OPTIMISATION

## ABSTRACT

The optimisation of neural networks can be sped up by orthogonalising the gradients before the optimisation step, ensuring the diversification of the learned representations. We orthogonalise the gradients of the layer's components/filters with respect to each other to separate out the intermediate representations. Our method of orthogonalisation allows the weights to be used more flexibly, in contrast to restricting the weights to an orthogonalised sub-space. We tested this method on ImageNet and CIFAR-10 resulting in a large decrease in learning time, and also obtain a speed-up on the semi-supervised learning BarlowTwins. We obtain similar accuracy to SGD without fine-tuning and better accuracy for naïvely chosen hyper-parameters.

## 1 INTRODUCTION

Neural network layers are made up of several identical, but differently parametrised, components, *e.g.* filters in a convolutional layer, or heads in a multi-headed attention layer. Layers consist of several components so that they can provide a diverse set of intermediary representations to the next layer, however, there is no constraint or bias, other than the implicit bias from the cost function, to learning different parametrisations. We introduce this diversification bias in the form of orthogonalised gradients and find a resultant speed-up in learning and sometimes improved performance, see fig. 1.

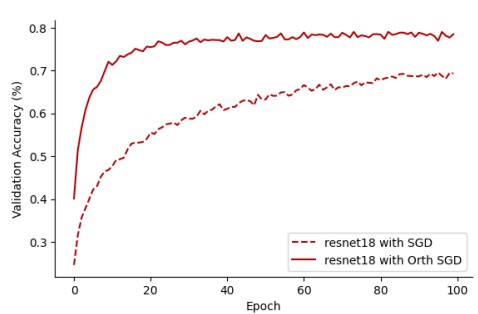

Figure 1: An example of the speed-up obtained by orthogonalising the gradients on CIFAR-10.

Our novel contributions include this new optimisation method, thorough testing on CIFAR-10 and ImageNet, additional testing on a semi-supervised learning method, and experiments to support our hypothesis.

In section 2 we detail the method and results to give an understanding of how this method works and its capabilities. Then, in section 3, we provide experimental justifications and supporting experiments for this method along with finer details of the implementation and limitations.

## 2 OVERVIEW OF NEW METHOD AND RESULTS

### 2.1 RELATED WORKS

Gradient orthogonalisation has been explored in the domain of multi-task learning (Yu et al., 2020) to keep the different tasks separate and relevant. However in this work we focus on orthogonalisation for improving single task performance.

Weight orthogonalisation has been extensively explored with both empirical (Bansal et al., 2018; Jia et al., 2017) and theoretical (Jia et al., 2019) justifications. However, modifying the weights during

training is unstable, and, in addition, it limits the weights to a tiny subspace. Deep learning is know to work well despite the immense size of the weight space, and as such we do not view this as an advantage. Xie et al. (2017) obtain improved performance over Stochastic Gradient Descent (SGD) via weight orthogonalisation and allows them to train very deep networks, we aim to achieve the same thing while being more flexible with model and optimisation method choice. We do this by orthogonalising the gradients before they are used by an optimisation method rather than modifying the weights themselves.

## 2.2 ORTHOGONALISING GRADIENTS

Given a neural network, $f$, with $L$ layers made from components, $c$,

$$f = \circ_{i=1}^{L} (f_i), \tag{1}$$
$$f_l(x) = [c_{l1}(x), c_{l2}(x), \ldots, c_{lN_l}(x)], \tag{2}$$

where $\circ$ is the composition operator, $N_l$ is the number of components in layer $l$, $c_l : \mathbb{R}^{S_{l-1} \times N_{l-1}} \to \mathbb{R}^{S_l}$ is a parametrised function and $c_{l_i}$ denotes $c_l$ parametrised with $\theta_{l_i} \in \mathbb{R}^{P_l}$ giving $f_l : \mathbb{R}^{S_{l-1} \times N_{l-1}} \to \mathbb{R}^{S_l \times N_l}$ parametrised by $\theta_l \in \mathbb{R}^{P_l \times N_l}$.

Let

$$G_l = [\nabla c_{l1}, \nabla c_{l2}, \ldots, \nabla c_{lN_l}], \tag{3}$$

be the $P_l \times N_l$ matrix of the components' gradients.

Then the nearest orthonormal matrix, *i.e.* the orthonormal matrix, $O_l$, that minimises the Frobenius norm of its difference from $G_l$

$$\min_{O_l} \|O_l - G_l\| \quad \text{subject to} \ \forall i, j : \langle O_{li}, O_{lj} \rangle = \delta_{ij},$$

where $\delta_{ij}$ is the Kronecker delta function, is the product of the left and right singular vector matricies from the Singular Value Decomposition (SVD) of $G_l$ (Trefethen & Bau III, 1997),

$$G_l = U_l \Sigma_l V_l^{\mathsf{T}}, \tag{4}$$
$$O_l = U_l V_l^{\mathsf{T}}. \tag{5}$$

Thus, we can adjust a first-order gradient descent method, such as Stochastic Gradient Descent with Momentum (SGDM) (Polyak, 1964), to make steps where the components are pushed in orthogonal directions,

$$v_l^{(t+1)} = \gamma v_l^{(t)} + \eta O_l^{(t)}, \ \text{and} \tag{6}$$
$$\theta_l^{(t+1)} = \theta_l^{(t)} - v_l^{(t+1)}, \tag{7}$$

where $v_l$ is the velocity matrix, $t \in \mathbb{Z}^{0+}$ is the time, $\gamma$ is the momentum decay term, and $\eta$ is the step size. We call this method Orthogonal Stochastic Gradient Descent with Momentum (Orthogonal-SGDM). This modification can clearly be applied to any first-order optimisation algorithm by replacing the gradients with $O_l^{(t)}$ before the calculation of the next iterate.

Code for creating orthogonal optimisers in PyTorch is provided at `https://anonymous.4open.science/r/Orthogonal-Optimisers`. And code for the experiments in this work is provided at `https://anonymous.4open.science/r/Orthogonalised-Gradients`

## 2.3 RESULTS

### 2.3.1 CIFAR-10

We trained a suite of models on the CIFAR-10 (Krizhevsky et al., 2009) data set with a mini-batch size of 1024, learning rate of $10^{-2}$, momentum of 0.9, and a weight decay of $5 \times 10^{-4}$ for 100 epochs. We then repeated this using Orthogonal-SGDM instead of SGDM and plot the results in figs. 2 and 3 and table 1.

Table 1: Test loss and accuracy across a suite of models on CIFAR-10 comparing normal SGDM with Orthogonal-SGDM, standard error across five runs.

| | Test Loss | | Test Accuracy (%) | |
| --- | --- | --- | --- | --- |
| | SGDM | Orthogonal-SGDM | SGDM | Orthogonal-SGDM |
| BasicCNN[1] | $0.7603 \pm 0.0061$ | $0.6808 \pm 0.0038$ | $73.60 \pm 0.19$ | $\mathbf{76.67} \pm 0.10$ |
| resnet20[2] | $0.6728 \pm 0.0301$ | $0.6766 \pm 0.0155$ | $79.14 \pm 0.62$ | $\mathbf{87.12} \pm 0.12$ |
| resnet44[2] | $0.7000 \pm 0.0166$ | $0.7600 \pm 0.0299$ | $79.81 \pm 0.37$ | $\mathbf{88.12} \pm 0.20$ |
| resnet18[3] | $0.9656 \pm 0.0104$ | $0.8427 \pm 0.0121$ | $77.01 \pm 0.21$ | $\mathbf{84.68} \pm 0.12$ |
| resnet34[3] | $1.0468 \pm 0.0134$ | $0.7087 \pm 0.0165$ | $75.86 \pm 0.26$ | $\mathbf{85.42} \pm 0.33$ |
| resnet50[3] | $1.2304 \pm 0.0462$ | $0.6797 \pm 0.0235$ | $67.99 \pm 0.73$ | $\mathbf{86.51} \pm 0.12$ |
| densenet121[3] | $1.0027 \pm 0.0132$ | $0.8669 \pm 0.0132$ | $75.26 \pm 0.30$ | $\mathbf{84.34} \pm 0.15$ |
| densenet161[3] | $1.1399 \pm 0.0096$ | $1.1688 \pm 0.1960$ | $75.81 \pm 0.20$ | $\mathbf{85.51} \pm 0.19$ |
| resnext50_32x4d[3] | $1.2470 \pm 0.0254$ | $0.6669 \pm 0.0223$ | $68.73 \pm 0.30$ | $\mathbf{86.37} \pm 0.24$ |
| wide_resnet50_2[3] | $1.4141 \pm 0.0337$ | $0.7018 \pm 0.0091$ | $69.42 \pm 0.33$ | $\mathbf{87.30} \pm 0.12$ |

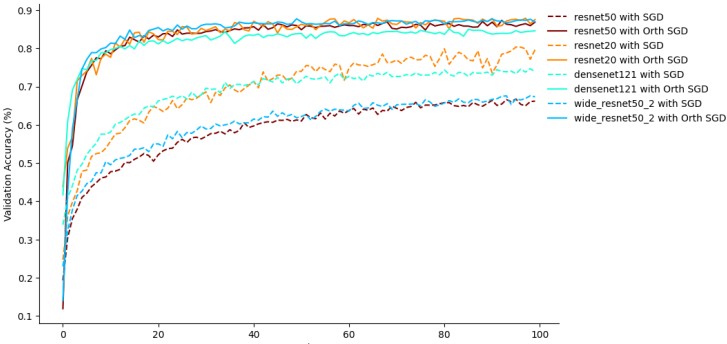

Figure 2: Validation accuracy from one run of SGDM vs Orthogonal-SGDM for a selection of models. Full plot in appendix C. Best viewed in colour.

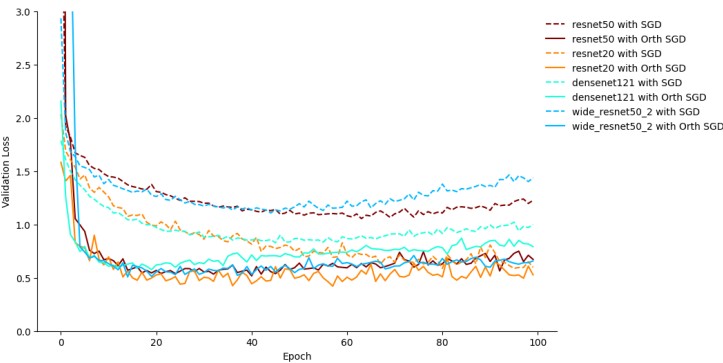

Figure 3: Validation losses from one run of SGDM vs Orthogonal-SGDM for a selection of models. Full plot in appendix C. Best viewed in colour.

Orthogonal-SGDM is more efficient and achieves better test accuracy than SGDM for every model we trained on CIFAR-10 without hyper-parameter tuning. The validation curves follow the training curves, figs. 4 and 5, and have the same patterns, this means that Orthogonal-SGDM exhibits the same generalisation performance as SGDM. More importantly though, we can see that the model

---

[1] As described in appendix B.1

[2] Model same as in He et al. (2015)

[3] From https://pytorch.org/vision/0.9/models.html

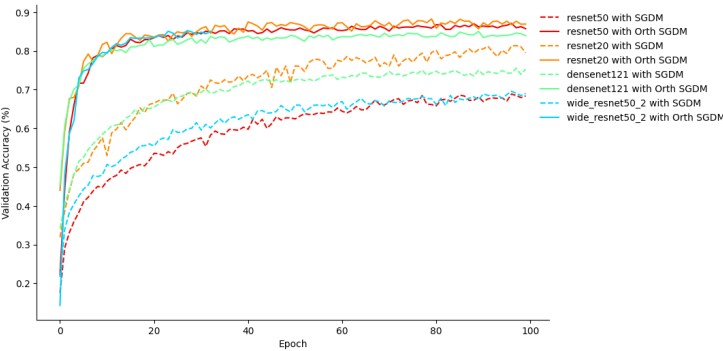

Figure 4: Train accuracy from one run of SGDM vs Orthogonal-SGDM for a selection of models. Best viewed in colour.

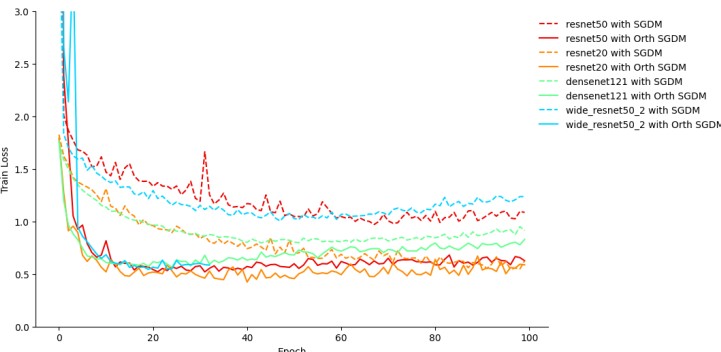

Figure 5: Train losses from one run of SGDM vs Orthogonal-SGDM for a selection of models. Best viewed in colour.

learns much faster at the beginning of training, as shown by fig. 2, this means that we do not need as many epochs to get to a well-performing network. This is especially good in light of the large data sets that new models are being trained on, where they are trained for only a few epochs, or even less (Brown et al., 2020).

For SGDM the performance of the residual networks designed for ImageNet (Deng et al., 2009) (18, 34, 50) get worse as the models get bigger. The original ResNet authors, He et al. (2015), note that unnecessarily large networks may over-fit on a small data set such as CIFAR-10. However, when trained with Orthogonal-SGDM, these models do not suffer from this over-parametrisation problem and even slightly improve in performance as the models get bigger, in clear contrast to SGDM. This agnosticism to over-parametrisation helps alleviate the need for the practitioner to tune a model's architecture to the task at hand to achieve a reasonable performance.

### 2.3.2 MATCHING RESNET'S PERFORMANCE

Having shown that Orthogonal-SGDM speeds up learning with non-optimised hyper-parameters, we now aim to show that it can achieve state-of-the-art results. To do this we use the same hyper-parameters as the original ResNet paper (He et al., 2015), which have been painstakingly tuned to benefit SGDM, to train using Orthogonal-SGDM.

This also tests the efficacy of Orthogonal-SGDM as a drop-in replacement for SGDM. Orthogonal-SGDM gets close to the original results, table 2, even though the hyper-parameters are perfected for SGDM. It is the authors' belief that with enough hyper-parameter tuning SGD or SGDM will be the best optimisation method; however, this experiment shows that Orthogonal-SGDM is robust to hyper-parameter choice and can easily replace SGDM in existing projects. Unfortunately, the authors do not have the compute-power to extensively hyper-parameter tune a residual network for

Table 2: Test loss and accuracy of a resnet20, as in He et al. (2015), on CIFAR-10; hyper-parameter tuned to normal SGDM vs Orthogonal-SGDM, standard error across five runs. Mini-batch size of 128, see section 3.6 for why this hyper-parameter value impedes Orthogonal-SGDM, learning-rate of 0.1, momentum of 0.9, weight-decay of $10^{-4}$, and a learning rate schedule of $\times 0.1$ at epochs 100, 150 for 200 epochs.

|  | Test Loss | Test Accuracy (%) |
| --- | --- | --- |
| SGDM (He et al., 2015) | — | **91.25** |
| SGDM | $0.4053 \pm 0.0054$ | $91.17 \pm 0.28$ |
| Orthogonal-SGDM | $0.4231 \pm 0.0043$ | $90.18 \pm 0.30$ |

Orthogonal-SGDM, however, it is exceedingly likely that better results would be achieved by doing so.

### 2.3.3 IMAGENET

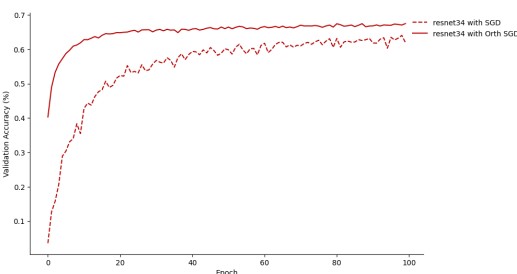

Figure 6: Validation accuracy of SGDM vs Orthogonal-SGDM on ImageNet

Orthogonal-SGDM also works on a large data set such as ImageNet (Deng et al., 2009) — fig. 6. Using a resnet34, mini-batch size of 1024, learning rate of $10^{-2}$, momentum of 0.9, and a weight decay of $5 \times 10^{-4}$, for 100 epochs. SGDM achieves a test accuracy of 61.9% and a test loss of 1.565 while Orthogonal-SGDM achieves 67.5% and 1.383 respectively. While these results are a way off the capabilities of the model they still demonstrate a significant speed-up and improvement from using Orthogonal-SGDM, especially at the start of learning, and further reinforces how a dearth of hyper-parameter tuning impedes performance.

### 2.3.4 BARLOW TWINS

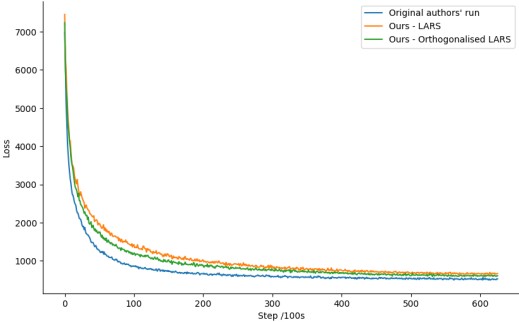

Figure 7: Barlow Twins loss during the unsupervised phase using LARS and Orthogonal LARS on ImageNet

Barlow Twins (Zbontar et al., 2021) is a semi-supervised method that uses "the cross-correlation matrix between the outputs of two identical networks fed with distorted versions of a sample" to avoid collapsing to trivial solutions. While the authors do provide code, we could not replicate their

results by running it. To train within our compute limitations we used a mini-batch size of 1024 instead of 2048 however this should not affect the results since "Barlow Twins does not require large batches" (Zbontar et al., 2021). Additionally, Barlow Twins uses the Layer-wise Adaptive Rate Scaling (LARS) algorithm (You et al., 2017), which is designed to adjust the learning rate based on the ratio between the magnitudes of the gradients and weights, there should be no significant slow-down, or speed-up, in learning due to the mini-batch size. We do not orthogonalise the gradients for the dense layers (see section 3.5).

Comparing our own runs, we establish that orthogonalising the gradients before the LARS algorithm does speed up learning as shown in fig. 7, in agreement with previous experiments. This is evidence that orthogonalising gradients is also beneficial for semi-supervised learning and, moreover, that optimisation algorithms other than SGDM can be improved in this way.

### 2.3.5 IN COMPARISON TO ADAM

We compare our method to the Adam optimiser (Kingma & Ba, 2014). Adam has found its place as a reliable optimiser that works over a wide variety of hyper-parameter sets, yielding consistent performance with little fine-tuning needed. We see our optimisation method occupying the same space as Adam.

Table 3: Test accuracy across a suite of hyper-parameter sets on CIFAR-10 on a resnet20, standard error across five runs. For Adam $\beta_2 = 0.99$.

| | | SGDM | | Adam | |
|---|---|---|---|---|---|
| LR | Momentum | Original | Orthogonal | Original | Orthogonal |
| $10^{-1}$ | 0.95 | $83.59_{\pm 2.09}$ | $\mathbf{85.58}_{\pm 0.98}$ | $38.95_{\pm 8.95}$ | $76.84_{\pm 1.53}$ |
| $10^{-2}$ | 0.95 | $82.66_{\pm 1.02}$ | $\mathbf{87.72}_{\pm 0.44}$ | $74.23_{\pm 2.38}$ | $86.48_{\pm 0.17}$ |
| $10^{-3}$ | 0.95 | $66.59_{\pm 0.44}$ | $\mathbf{85.88}_{\pm 0.33}$ | $83.08_{\pm 0.76}$ | $85.12_{\pm 0.06}$ |
| $10^{-1}$ | 0.9 | $82.52_{\pm 1.16}$ | $\mathbf{85.06}_{\pm 0.47}$ | $28.26_{\pm 7.16}$ | $73.62_{\pm 2.96}$ |
| $10^{-2}$ | 0.9 | $79.96_{\pm 0.48}$ | $\mathbf{87.44}_{\pm 0.25}$ | $73.46_{\pm 1.19}$ | $85.26_{\pm 0.38}$ |
| $10^{-3}$ | 0.9 | $60.69_{\pm 0.18}$ | $84.67_{\pm 0.21}$ | $83.16_{\pm 0.66}$ | $\mathbf{85.25}_{\pm 0.31}$ |
| $10^{-1}$ | 0.8 | $84.16_{\pm 0.43}$ | $\mathbf{86.01}_{\pm 0.74}$ | $27.50_{\pm 6.76}$ | $71.88_{\pm 4.25}$ |
| $10^{-2}$ | 0.8 | $77.42_{\pm 0.98}$ | $\mathbf{87.18}_{\pm 0.12}$ | $72.60_{\pm 1.76}$ | $86.75_{\pm 0.26}$ |
| $10^{-3}$ | 0.8 | $53.21_{\pm 0.43}$ | $82.95_{\pm 0.40}$ | $80.89_{\pm 1.93}$ | $\mathbf{85.52}_{\pm 0.29}$ |
| $10^{-1}$ | 0.5 | $80.08_{\pm 0.36}$ | $\mathbf{87.37}_{\pm 0.18}$ | $18.39_{\pm 4.84}$ | $72.10_{\pm 2.83}$ |
| $10^{-2}$ | 0.5 | $68.64_{\pm 1.05}$ | $\mathbf{86.05}_{\pm 0.10}$ | $71.62_{\pm 1.95}$ | $84.22_{\pm 0.69}$ |
| $10^{-3}$ | 0.5 | $43.51_{\pm 1.02}$ | $78.68_{\pm 0.77}$ | $81.67_{\pm 1.05}$ | $\mathbf{84.21}_{\pm 0.43}$ |

Our method outperforms Adam on all but one hyper-parameter set — table 3. In addition since we can apply our method to any previous optimisation method, we also test Orthogonal-Adam and find that it outperforms Adam too including at high learning rates where Adam suffers from blow-ups. See figs. 13 and 14 for the training plots.

## 3 DISCUSSION OF PROBLEM AND METHOD

### 3.1 NORMALISATION

When we perform SVD on the reshaped gradient tensor, we obtain an orthonormal matrix, since this changes the magnitude of the gradient we look at the effect of this normalisation. Normalised SGDM (N-SGDM) (Nesterov, 2003) provides an improvement in non-convex optimisation since it is difficult to get stuck in a local minimum as the step size is not dependent on the gradient magnitude. However, it hinders convergence to a global minimum since there is no way of shortening the step size; deep learning is highly non-convex and is unlikely to be optimised to a global minimum. Therefore, it stands to reason that normalising the gradient would speed up the optimisation of deep networks.

We compare N-SGDM to normalising the gradients per component — *i.e.* normalising the columns of $G_l$, eq. (3), instead of orthognormalising it — Component Normalised SGDM (CN-SGDM), as well as to SGDM and Orthogonal-SGDM. N-SGDM improves over SGDM, and CN-SGDM improves over N-SGDM except from the oft case where it diverges. Finally, Orthogonal-SGDM obtains the best solutions while remaining stable on all the models.

Table 4: Test accuracy for several models trained with SGDM, Normalised SGDM, Component Normalised SGDM, and Orthogonal-SGDM; trained as in section 2.3.1.

|  | SGDM | N-SGDM | CN-SGDM | Orthogonal-SGDM |
|---|---|---|---|---|
| BasicCNN | $73.68_{\pm 0.27}$ | $73.72_{\pm 0.45}$ | $74.53_{\pm 0.32}$ | $\mathbf{76.75}_{\pm 0.23}$ |
| resnet18 | $76.83_{\pm 0.22}$ | $78.94_{\pm 0.19}$ | $0.00_{\pm 0.00}$ | $\mathbf{84.94}_{\pm 0.10}$ |
| resnet50 | $69.35_{\pm 0.30}$ | $79.35_{\pm 0.21}$ | $0.00_{\pm 0.00}$ | $\mathbf{86.59}_{\pm 0.10}$ |
| resnet44 | $79.73_{\pm 1.27}$ | $83.60_{\pm 0.77}$ | $84.44_{\pm 0.55}$ | $\mathbf{87.49}_{\pm 0.39}$ |
| densenet121 | $75.45_{\pm 0.20}$ | $79.06_{\pm 0.04}$ | $0.00_{\pm 0.00}$ | $\mathbf{84.86}_{\pm 0.07}$ |

## 3.2 DIVERSIFIED INTERMEDIARY REPRESENTATIONS

Along with different parametrisations we also desire different intermediary representations, a model will perform better if its layers output $N$ different representations as opposed to $N$ similar ones.

Given $x_l$ are the resulting representations from the intermediary layers,

$$x_l = \left( \circ_{i=1}^l f_i \right)(x_0)$$

where $x_0$ is the input and $x_l$ is the intermediary representation after layer $l$. Then $x_{l_i}$ is the representation provided by $c_{l_i}$.

We now look at the statistics of the absolute cosine of all distinct pairs of different latent features,

$$R_l = \left\{ \left| \langle x_{l_i}, x_{l_j} \rangle_2 \right| \mid i < j \right\}.$$

The representations have smaller cosines when using Orthogonal-SGDM versus SGDM — fig. 8. In addition, Orthogonal-SGDM shows a steady decline in cosine similarity throughout training. This indicates that more information is being passed to the next layer as the network is trained.

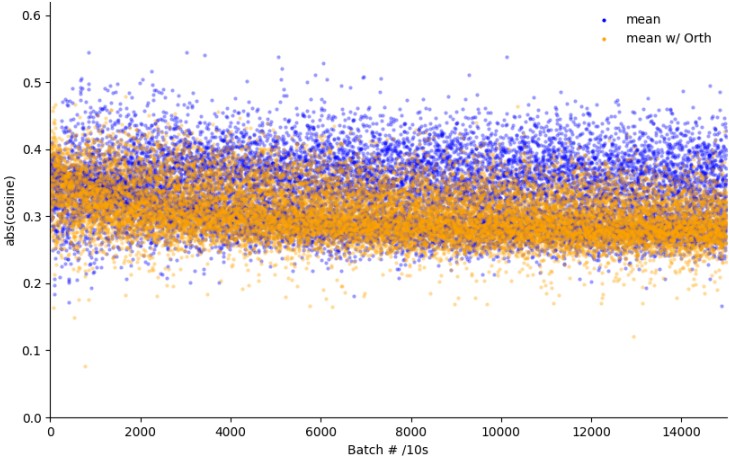

Figure 8: Mean of the absolute cosine of all distinct pairs of different intermediary representations, $\mathbb{E}[R_l]$, $l \in \{1, 2, 3\}$, for all layers of a BasicCNN trained on CIFAR-10 as in section 2.3.1.

## 3.3 DEAD PARAMETERS

Dead parameters occur when the activation function has a part with zero gradient, *e.g.* a Rectified Linear Unit (ReLU). If the result of the activation remains in this part, then the gradients of the

preceding parameters will be zero and prevented from learning. This limits the model's capacity based on a parameterisation, however temporarily dead parameters can be beneficial and act as a regulariser, similar to dropout. To detect temporarily dead parameters, we simply look for parameters with zero gradient. Comparing the amount of dead parameters produced by SGDM versus Orthogonal-SGDM, figs. 9a and 9b respectively, shows that Orthogonal-SGDM ends with around and order of magnitude more temporarily dead parameters. This implies a much higher regularisation which helps to explain Orthogonal-SGDM's insensitivity to over-parametrisation.

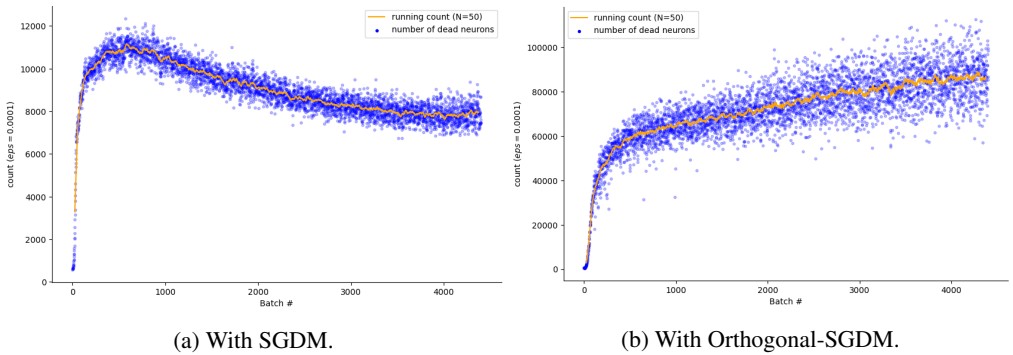

(a) With SGDM.                    (b) With Orthogonal-SGDM.

Figure 9: Number of temporarily dead parameters in layer2[1].conv2 of a resnet50 trained as in section 2.3.1.

### 3.4 IMPLEMENTATION DETAILS

While the QR decomposition is the most common orthogonalisation method, it is, in practice, less stable as the gradients are rank deficient (Demmel, 1997, Section 3.5), *i.e.* they have at least one small singular value. Orthogonal-SGDM has a longer wall time than SGDM because of the added expense of the SVD which has non-linear time complexity in the matrix size. In practice, we have found that the calculation of the SVD is either more than made up for by the speed up in iterates or a prohibitively expensive cost, with dense layers being the largest and so most problematic.

While there exist methods for computing an approximate SVD which are faster, we have used PyTorch's default implementation since we are more concerned with Orthogonal-SGDM's performance and efficiency in iterates and not in wall time. Even so the overhead is small, training a resnet20 as in section 2.3.1 takes 720.3 seconds with 96.4 of them taken up by the SVD calculation — an increase of 15.5% over normal SGDM. While this is a significant amount of time we can see that our method can take only 2% of the number of epochs to reach the same accuracy — fig. 2.

It is doubtful that convergence of SVD is needed, so a custom matrix orthogonalisation algorithm, that has the required stability but remains fast and approximate, will reduce the computation overhead significantly and may allow previously infeasible networks to be optimised using Orthogonal-SGDM. However, we note that even with a more suitable implementation, this method would still bias towards many smaller layers for a deeper, thinner network.

### 3.5 FULLY CONNECTED LAYERS

Fully connected or dense layers also fit our component model from eq. (2) where the components are based on the inner product of the input and the parametrisation,

$$c_{l_i}(x) = \sigma(\langle \text{flatten}(x), \theta_{l_i} \rangle),$$

where $\sigma$ is an activation function, $S_l = 1$ giving $f_l : \mathbb{R}^{S_{l-1} \times N_{l-1}} \rightarrow \mathbb{R}^{N_l}$ and $\theta_l \in \mathbb{R}^{S_{l-1} \cdot N_{l-1} \times N_l}$ as desired. Intuitively, each column of the weight matrix acts as a linear map resulting in one item in the output vector. Thus,

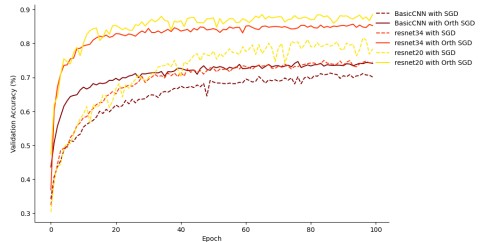

Figure 10: Orthogonalising just the convolutional filters vs both the convolutional layers and final dense layer on CIFAR-10; trained as in section 2.3.1.

the gradients of fully connected layers can also be orthogonalised.

Table 5: Test accuracy and loss for Orthogonal-SGDM on CIFAR-10 when orthogonalising all layers vs orthogonalising just the convolutional layers. Trained as in section 2.3.1, standard error across five runs.

|  | SGDM | | Orthogonal-SGDM | | Conv Orthogonal-SGDM | |
| --- | --- | --- | --- | --- | --- | --- |
|  | Loss | Acc (%) | Loss | Acc (%) | Loss | Acc (%) |
| BasicCNN | $0.7603 \pm 0.0061$ | $73.60 \pm 0.19$ | $0.6808 \pm 0.0038$ | $76.67 \pm 0.10$ | $0.6732 \pm 0.0041$ | $\mathbf{76.80} \pm 0.18$ |
| resnet34 | $1.0468 \pm 0.0134$ | $75.86 \pm 0.26$ | $0.7087 \pm 0.0165$ | $85.42 \pm 0.33$ | $0.6268 \pm 0.0105$ | $\mathbf{85.68} \pm 0.21$ |
| resnet20 | $0.6728 \pm 0.0301$ | $79.14 \pm 0.62$ | $0.6766 \pm 0.0155$ | $87.12 \pm 0.12$ | $0.4824 \pm 0.0225$ | $\mathbf{87.70} \pm 0.40$ |

As noted in section 3.4 the extra wall time is dominated by the largest parameter, this is often the dense layer; table 5 shows that for CIFAR-10 there is no impact on the error rate from not orthogonalising the final dense layer, and the training curves are the same shape — fig. 10. While both the error rates and losses decrease when not orthogonalising the dense layer we hesitate to say that orthogonalising dense layers is detrimental since these networks only have a dense final classification layer which is qualitatively different from intermediary dense layers.

### 3.6 LIMITATIONS DUE TO MINI-BATCH SIZE

Orthogonal-SGDM does not perform as well as SGDM when the mini-batch size is extremely small, fig. 11, due to the increased levels of noise for the SVD. This is the most likely reason that the resnet20 from section 2.3.2 fails to match the original performance.

A mini-batch size of 16 is where Orthogonal-SGDM starts to outperform SGDM on a resnet18 for CIFAR-10. Few models need such small mini-batch sizes, but if they do then SGDM would be a more suitable optimisation algorithm. In addition to the learning collapse, the time taken by SVD is only dependent on the parameter size and not the mini-batch size, so increasing the number of mini-batches per epoch also increases the wall time to train. The reason for the collapse in training with small mini-batch sizes will be subject to further research.

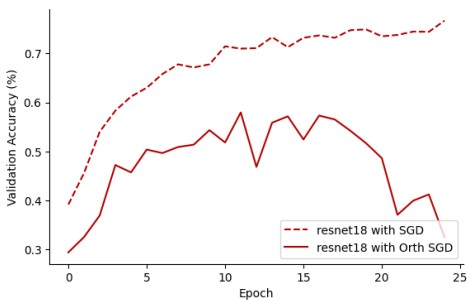

Figure 11: CIFAR-10 with mini-batch size=4 trained as in section 2.3.1.

## 4 CONCLUSION

In this work we have laid out a new optimisation method, tested it on different models and data sets, showing close to state-of-the-art results out of the box and robustness to hyper-parameter choice and over-parametrised models. Orthogonal-SGDM also has practical application in problems such as object detection and semantic segmentation since they make use of a pre-trained image classification backbone.

SGDM with a vast amount of hyper-parameter tuning still reigns supreme, but Orthogonal-SGDM is an excellent method for quick verification of models or for prototyping — when we want decent results fast, but do not need the absolute best performing model. However, as more data set sizes are growing more models are being trained on fewer to less than one epoch of data (Brown et al., 2020) leading to an extremely limited ability to tune the hyper-parameters.

Lastly, we mentioned briefly in section 1 how attention heads fit our model but, since they are beyond the scope of this work, we will explore the potential gain in using Orthogonal-SGDM with them in future work, and expect a similarly exceptional gain will be obtained.

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

# A  COSINE THRESHOLD

We use the cosine metric

$$\langle x, y \rangle_2 = \frac{x \cdot y}{\|x\|_2 \|y\|_2} \tag{8}$$

in this work since it allows the comparison of the directions of high-dimensional vectors, and so obtain insight about the surface we are optimising on. However, we note that the "significance" of the cosine between two random vectors depends on their size.

Assuming both vectors' components are random variables

$$x = [x_1, x_2, \ldots, x_N],$$

where $x_i \sim \mathcal{N}(0, \sigma^2)$, then the components of their dot product

$$\langle x, y \rangle = [x_1 \cdot y_1, x_2 \cdot y_2, \ldots x_N \cdot y_N],$$

have variance $\sigma^4$. Now, from the central limit theorem, the dot product has variance $\frac{\sigma^4}{N}$ where $N$ is the size of the vectors. Finally, dividing by the magnitude of the vectors gives a variance of $\bar{\sigma} = \frac{\sigma^2}{N}$ for the cosine metric.

To gain some understanding of the significance of a cosine distance, we define a four-sigma threshold on the distribution of cosines, so, assuming $\sigma = 1, \mu = 0$, we get a threshold value of

$$\mu \pm 4\bar{\sigma} = 0 \pm \frac{4\sigma}{\sqrt{N}}$$

$$= \pm \frac{4}{\sqrt{N}} \tag{9}$$

This is important because a distance of $0.1$ might seem small, but for $10,000$-dimensional vectors, it easily clears our significance threshold.

# B  MODEL SUMMARIES

## B.1  BASICCNN

```
        Layer (type)              Output Shape          Param #
================================================================
          Conv2d-1            [-1, 32, 16, 16]              896
     BatchNorm2d-2            [-1, 32, 16, 16]               64
          Conv2d-3              [-1, 32, 8, 8]            9,248
     BatchNorm2d-4              [-1, 32, 8, 8]               64
          Conv2d-5              [-1, 32, 4, 4]            9,248
     BatchNorm2d-6              [-1, 32, 4, 4]               64
        Linear-7                     [-1, 10]            5,130
      BasicCNN-8                     [-1, 10]                0
================================================================
Total params: 24,714
Trainable params: 24,714
Non-trainable params: 0
----------------------------------------------------------------
Input size (MB): 0.01
Forward/backward pass size (MB): 0.16
Params size (MB): 0.09
Estimated Total Size (MB): 0.27
----------------------------------------------------------------
```

## C   FULL RESULTS PLOT

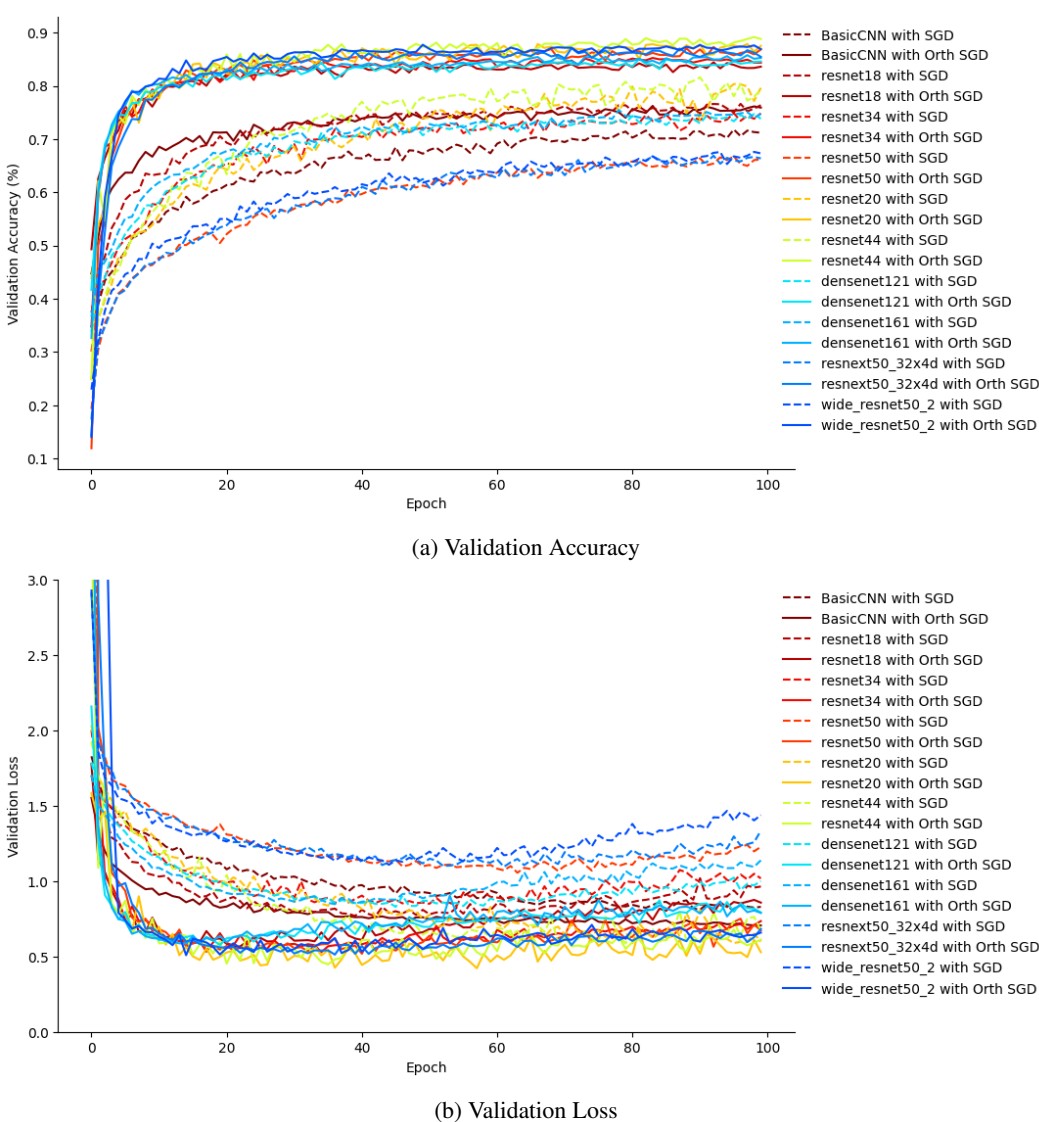

(a) Validation Accuracy

(b) Validation Loss

Figure 12: SGDM vs Orthogonal SGDM

Table 6: Test loss across a suite of hyper-parameter sets on CIFAR-10 on a resnet20, standard error across five runs. For Adam $\beta_2 = 0.99$.

| | | SGDM | | Adam | |
|---|---|---|---|---|---|
| LR | Momentum | Original | Orthogonal | Original | Orthogonal |
| $10^{-1}$ | 0.95 | $0.5487_{\pm 0.0913}$ | $\mathbf{0.4916}_{\pm 0.0420}$ | $2.4148_{\pm 0.7090}$ | $0.7938_{\pm 0.0663}$ |
| $10^{-2}$ | 0.95 | $0.5762_{\pm 0.0431}$ | $\mathbf{0.4772}_{\pm 0.0277}$ | $0.8395_{\pm 0.1079}$ | $0.5415_{\pm 0.0153}$ |
| $10^{-3}$ | 0.95 | $0.9323_{\pm 0.0149}$ | $\mathbf{0.4988}_{\pm 0.0114}$ | $0.6013_{\pm 0.0305}$ | $0.5998_{\pm 0.0131}$ |
| $10^{-1}$ | 0.9 | $0.6089_{\pm 0.0535}$ | $\mathbf{0.5370}_{\pm 0.0230}$ | $9.0193_{\pm 5.9607}$ | $0.9803_{\pm 0.1607}$ |
| $10^{-2}$ | 0.9 | $0.6217_{\pm 0.0194}$ | $\mathbf{0.4841}_{\pm 0.0087}$ | $0.8898_{\pm 0.0447}$ | $0.5903_{\pm 0.0201}$ |
| $10^{-3}$ | 0.9 | $1.0977_{\pm 0.0036}$ | $\mathbf{0.5042}_{\pm 0.0055}$ | $0.6045_{\pm 0.0312}$ | $0.5829_{\pm 0.0102}$ |
| $10^{-1}$ | 0.8 | $\mathbf{0.5395}_{\pm 0.0091}$ | $0.5414_{\pm 0.0459}$ | $4.5143_{\pm 1.6628}$ | $1.0984_{\pm 0.2710}$ |
| $10^{-2}$ | 0.8 | $0.6669_{\pm 0.0290}$ | $\mathbf{0.4940}_{\pm 0.0163}$ | $0.8741_{\pm 0.0627}$ | $0.5114_{\pm 0.0204}$ |
| $10^{-3}$ | 0.8 | $1.2805_{\pm 0.0106}$ | $\mathbf{0.5238}_{\pm 0.0034}$ | $0.6950_{\pm 0.0656}$ | $0.5671_{\pm 0.0088}$ |
| $10^{-1}$ | 0.5 | $0.6796_{\pm 0.0158}$ | $\mathbf{0.5003}_{\pm 0.0148}$ | $15683.8291_{\pm 15681.5332}$ | $0.9481_{\pm 0.1072}$ |
| $10^{-2}$ | 0.5 | $0.8927_{\pm 0.0304}$ | $\mathbf{0.4950}_{\pm 0.0121}$ | $0.9594_{\pm 0.0846}$ | $0.6610_{\pm 0.0285}$ |
| $10^{-3}$ | 0.5 | $1.5309_{\pm 0.0189}$ | $0.6284_{\pm 0.0186}$ | $0.6482_{\pm 0.0511}$ | $\mathbf{0.6228}_{\pm 0.0213}$ |

Table 7: Test loss for several models trained with SGDM, Normalised SGDM, Component Normalised SGDM, and Orthogonal-SGDM.

| | SGDM | N-SGDM | CN-SGDM | Orthogonal-SGDM |
|---|---|---|---|---|
| BasicCNN | $0.7559_{\pm 0.0065}$ | $0.7637_{\pm 0.0098}$ | $0.7443_{\pm 0.0094}$ | $\mathbf{0.6824}_{\pm 0.0081}$ |
| resnet18 | $0.9252_{\pm 0.0098}$ | $0.9726_{\pm 0.0214}$ | $\text{nan}_{\pm \text{nan}}$ | $\mathbf{0.7938}_{\pm 0.0083}$ |
| resnet50 | $1.0950_{\pm 0.0181}$ | $0.9454_{\pm 0.0126}$ | $\text{nan}_{\pm \text{nan}}$ | $\mathbf{0.6785}_{\pm 0.0076}$ |
| resnet44 | $\mathbf{0.7093}_{\pm 0.0678}$ | $0.7641_{\pm 0.0441}$ | $0.7902_{\pm 0.0477}$ | $0.7694_{\pm 0.0426}$ |
| densenet121 | $0.9357_{\pm 0.0071}$ | $1.0096_{\pm 0.0167}$ | $\text{nan}_{\pm \text{nan}}$ | $\mathbf{0.8142}_{\pm 0.0084}$ |

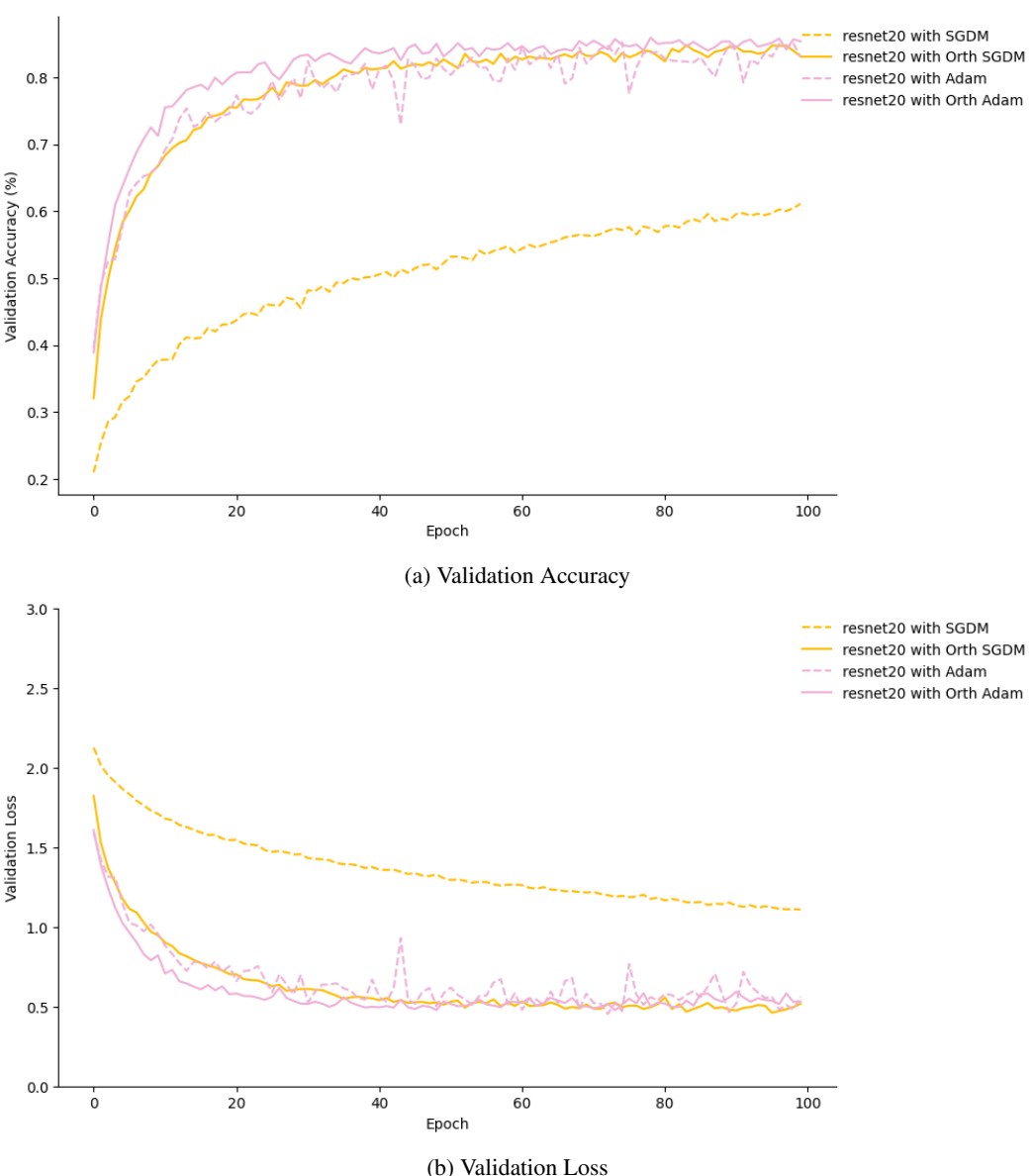

(a) Validation Accuracy

(b) Validation Loss

Figure 13: A compassion of Adam and SGDM, learning rate = $1 \times 10^{-3}$, $\beta_1 = 0.9$, $\beta_2 = 0.99$.

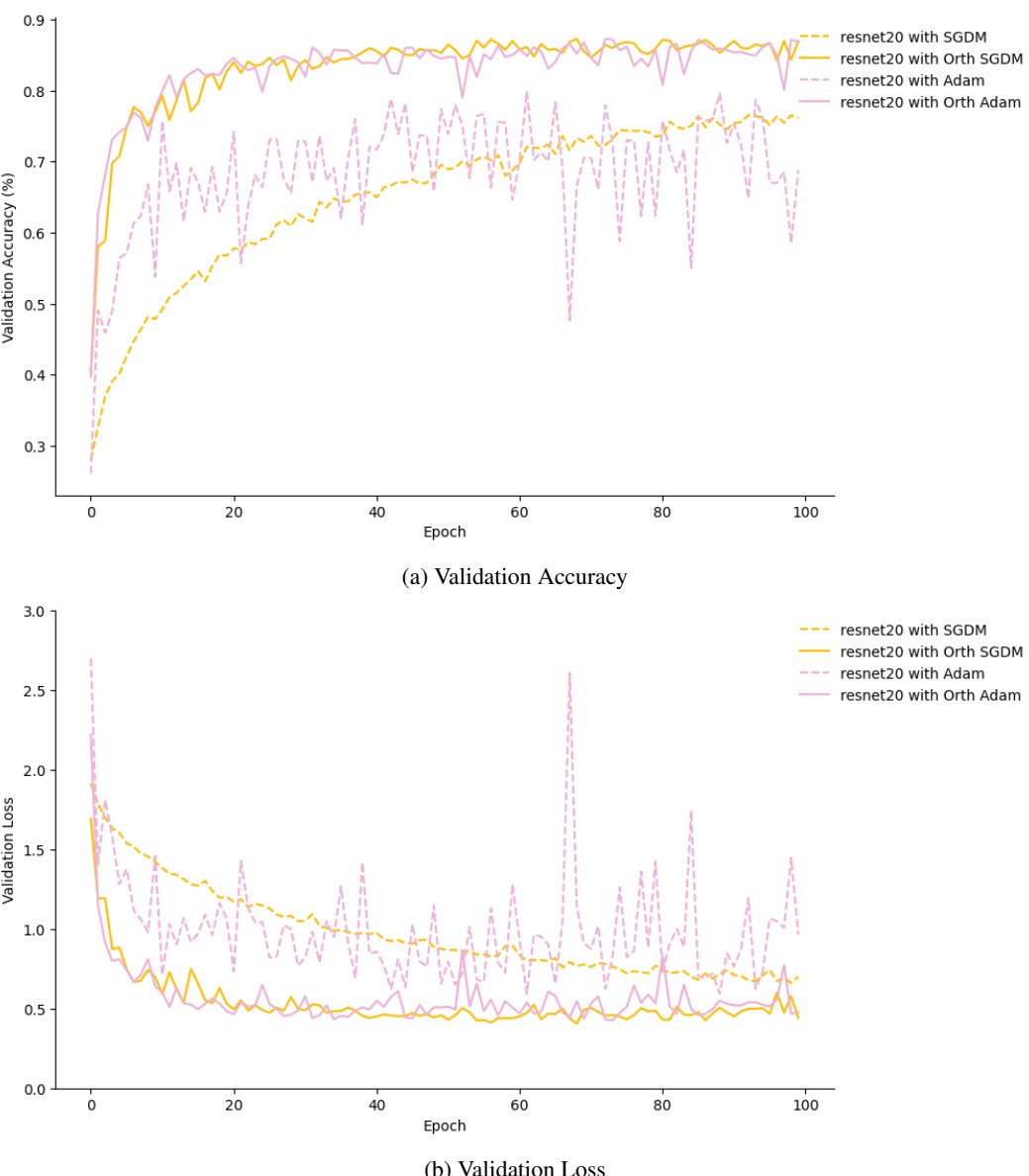

(a) Validation Accuracy

(b) Validation Loss

Figure 14: A compassion of Adam and SGDM, learning rate $= 1 \times 10^{-2}$, $\beta_1 = 0.8$, $\beta_2 = 0.99$.

