# OpenReview forum: "Orthogonalising gradients to speedup neural network optimisation"
_ICLR.cc/2022/Conference — ICLR 2022 Submitted_

### Official Review · Reviewer_Pgpr · 2021-10-30

**Correctness:** 2
**Technical Novelty And Significance:** 2
**Empirical Novelty And Significance:** 2
**Recommendation:** 3
**Confidence:** 4

**Main Review:**

The conducted experiments do not support the author's main claim of improved training speed. This should be measured on the training loss. Therefore all presented Figures can only support claims dealing with generalization error - is there a reason for not showing the training loss curves? I find this irritating.

Furthermore, the motivation of the paper that the layers representations/layers weights start aligning at the beginning of training seems not true and leads to the authors questioning their own hypthoses - see Discussion 3. I find this irritating as well.

I am recommending a reject.

Nevertheless, the proposed idea seems interesting and I strongly urge the authors to study neural network optimization papers that look into the convergence speed of deep networks. Here, usually, experiments are not (mainly) done on very large networks / datasets and so the authors might have an easier time investigating the proposed method under limited compute (since they mention this limitation).
See for example the experiments conducted in https://arxiv.org/pdf/1503.05671.pdf.

The authors should tune the method they compare against as well as their own method since hyperparameters are crucial for training deep nets - since this is not done thoroughly in the paper, interpretations of the conducted experiments are difficult.


**Summary Of The Paper:**

The paper claims that neural network training leads especially at the beginning of training to similar representations across the network.  To tackle this, they propose to orthogonalize gradients across the layer-wise feature dimensions of deep neural networks.

**Summary Of The Review:**

Experiments seem not to align with questions asked by the authors. Technical quality is not sufficient. Reject.

---

> ### Author Response · Authors · 2021-11-22
> **Response to review**
>
> Thank you for taking the time to review our paper.
>
> Training curves and discussion has been added to the paper.
> While the training curve is the direct result of the optimisation the validation accuracy is what we actually want to increase --- that is, we want to have good generalised performance.
> We mainly use the validation curves since looking that the training curves will not tell us if the method will hold up in practice as the generalisation error might be huge, and it is exceedingly unlikely that any method can optimise the validation set and not the training set.
>
> We agree that our initial motivation is unhelpful to the communication of our method and have now removed it.
>
> We have added a section of hyper-parameter sets for the resnet20 model to show the results are constant.

---

> > ### Comment · Reviewer_Pgpr · 2021-11-23
> > **Thanks**
> >
> > Thanks for the comments and improvements. Although validation set accuracy is what many people are interested in, training set accuracy should be the focus of this paper and is a very important problem on its own.
> >
> > I hope the method will be thoroughly investigated and submitted again to another conference.

---

### Official Review · Reviewer_S4zJ · 2021-11-02

**Correctness:** 2
**Technical Novelty And Significance:** 2
**Empirical Novelty And Significance:** 2
**Recommendation:** 3
**Confidence:** 5

**Main Review:**

*Summary*

I find the research presented in this work interesting and I appreciate the simplicity in the method provided. Idea of orthogonalizing parameters (a) at initialization [1] (b) during training [2] and (c) after training [3] is widely studied, however I am not aware of any work proposing what is being proposed in this paper. Even though the idea is possibly novel, the assumptions, related work and experiments lack the depth needed for such experimental work to be useful for future research.

*Weaknesses*
- (1) ResNet34 baseline should get something around 73% on ImageNet. Similarly I found CIFAR accuracies to be very low. ResNet-20 with batch normalization should get 92-93% on Cifar10. I recommend authors to use strong baselines, otherwise it is very difficult to convince the reader whether this method is useful in practice.
- (2) Authors should share training curves and discuss whether why orthogonalization helps training dynamics or generalization (or both).
- (3) Would using ortogonal regularization and orthogonal initilization fix the problem? As explained briefly in the summary, orthogonality is widely studied in neural networks. Authors need to discuss and compare their methods against previous methods.
- (4) Cost of motivated method needs to be discussed. Do we need to calculate SVD at every iteration. Can we start from previous bases? How does the cost and the gains compare with orthogonal initialization and regularization?
- (5) Orthogonal basis is normalized, which affects the magnitude of the gradients; therefore learning rate should be searched for SGD and proposed method separately using a validation set. If computational resources are scarce I recommend authors to focus on one of the resnets in cifar (instead of trying 5 different depths).
- (6) "...for Orthogonal-SGDM we see that the components’ parameters are more similar than they were with SGDM and even include the initial spike. The reason behind this counter-intuitive result will be studied in the future" This is the key motivation for provided method, which seems to be not-true. I recommend authors to revisit/update their assumptions and hypothesis in the next iteration.


*Minor*
- can be sped up
- We obtain similar accuracy to SGD without fine-tuning (not sure what this refers to).
- component->neuron.
- The notation in 2.2 can be simplifed drastically. I recommend authors to assume fully connected layer and discuss convolutional layers in appendix.
- dearth
- table 2 -> Table 2
- Figure 7 and 8 needs to be summarized like it is done in figure-6. For example seaborn.lineplot would be a good fit.



1) https://arxiv.org/pdf/2001.05992.pdf
2) https://arxiv.org/pdf/1905.05929.pdf
3) https://arxiv.org/abs/1404.0736

## After Rebuttal
I read authors response and most of the limitations (1,3,5) are still a concern, thus I keep my score and encourage authors to address them in the future revisions.

**Summary Of The Paper:**

This paper proposes to orthogonalize the gradients of each neuron (within a layer) in order to improve training dynamics. They build on the assumption that neurons in a layer tend to learn similar features early in the training, and thus pushing them in orthogonal directions would yield to better results. The method requires calculating SVD on gradients at every iteration, thus unlikely to be practical (no FLOPs/run-time discussion). Training curves on Cifar10 and Imagenet shows faster convergence and better generalization.

**Summary Of The Review:**

Even though the idea of orthogonalization of the gradients is interesting, this work lacks the comparison to prior art and proper baselines. I recommend authors to work with proper baselines and compare their method against other orthogonalization techniques like orthogonal initialization and regularization. Furthermore the cost of doing SVD at every training iteration needs to be justified and discussed in detail.

---

> ### Author Response · Authors · 2021-11-22
> **Response to review**
>
> Thank you for taking the time to review our paper.
>
> The CIFAR-10 baselines are from using SGD with "default" hyper-parameters, while we concede there is no set of parameters that work for every problem we have chosen a reasonable set that many practitioners would start off at.
> The high-performing hyper-parameter sets are gained through a thorough search to show-case the model's capacity, and are not wholly representative of how that optimisation would work applied to another task.
> Additionally, see our new section on a comparison to Adam.
>
> Training curves and discussion have been added to the paper.
>
> We have added a section on specific timings for the SVD and show that while it is not insignificant, it is more than outweighed by the speed-up obtained.
>
> The fact that the learning rate is not as sensitive with our method is an advantage, see the added results with full normalisation and normalisation as it occurs when orthogonalising the gradients with SVD.
> We have also included results for a resnet20 with different hyper-parameter sets
>
> We agree that our initial motivation is unhelpful to the communication of our method and have now removed it.

---

### Official Review · Reviewer_CSRa · 2021-11-02

**Correctness:** 2
**Technical Novelty And Significance:** 2
**Empirical Novelty And Significance:** 2
**Recommendation:** 3
**Confidence:** 4

**Main Review:**

Strengths
=========
* Orthogonalising gradients, rather than weights, seems to be a novel idea.

Weaknesses
==========
* There is no explanation for how the authors are computing the SVD of convolutional layers. Is the doubly block circulant matrix that corresponds to the gradient of the convolutional layer explicitly constructed and then decomposed? Or are the gradients heuristically "reshaped" into a matrix and a quasi-SVD performed? If the former, how can this be done efficiently? If the latter, is orthogonality still actually enforced?
* The performance comparisons are lacking. Picking one set of hyperparameters is not a sufficient procedure for comparing two algorithms. It could be the case that the particular set of hyperparameters chosen are well-suited to Orthogonal-SGDM, but maybe there exists another set of hyperparameters that allows SGDM to achieve superior performance. Given the unusually low performance of the different ResNet models considered in the experiments, this seems like a very real possibility.
* The motivation for introducing this method is contradicted by the experimental results, so it is unclear why/when one might see benefits from using this approach. In particular, the authors state that each randomly initialised unit in a layer will converge towards the same weights near the beginning of training. I do not really follow why this would be the case, and the experiments in Section 3 seem to directly contradict this.
* Discussion and empirical comparison to show how this is different to other orthogonalisation and projection-based optimisation/regularisation approaches is needed.

Post-rebuttal
==========
Thank you for clarifying the point about how this is implemented for convolutional layers. Thinking about this a bit more, I suspect both options I proposed are actually equivalent, which is an interesting property of your method you may wish to prove/highlight!

However, many of my concerns remain:
* The performance of the baselines is still very low compared to number typically seen in the literature for similar experimental setups, which indicates that something is either missing from the presentation (to explain why they are so different), or that there is a technical problem with the experiments.
* It's good that the authors have removed the faulty motivation, but that only partially addresses my criticism: it's not obvious why orthogonalising the gradients is something we should do.
* Perhaps I should have been more clear: I would like to see discussion about *why* orthogonalising gradients will have a different effect to orthogonalising weights.

**Summary Of The Paper:**

The paper proposes inserting a gradient orthogonalisation step before each update for first-order optimisation methods. The orthogonalisation is accomplished via SVD, and the reported results show that in some settings this can lead to significant differences in performance and convergence rate compared to first-order optimisers that do not have the orthogonalisation.

**Summary Of The Review:**

The work proposes a novel idea, but the presentation, evaluation, and analysis is not yet sufficient to warrant publication.

---

> ### Author Response · Authors · 2021-11-22
> **Response to review**
>
> Thank you for taking the time to review our paper.
>
> The gradients are reshaped into a matrix on which the SVD is performed, and then reshaped back into the gradient's original shape.
> Orthogonality is enforced between the filter vectors.
>
> We have expanded the hyper-parameter set for resnet20 to include multiple sets where Orthogonal-SGDM still over-performs SGDM.
> Additionally, see our new section on a comparison to Adam.
>
> We agree that our original motivation is not helpful to the paper, and have subsequently removed it.
>
> Previous orthogonalisation based methods directly modify the weights, whereas our method adjust the inputs to existing optimisation methods.

---

### Official Review · Reviewer_snPa · 2021-11-03

**Correctness:** 2
**Technical Novelty And Significance:** 2
**Empirical Novelty And Significance:** 2
**Recommendation:** 3
**Confidence:** 5

**Main Review:**

The paper touches upon orthogonalization of gradients, which is actually something that has already been explored in the literature - see for example https://arxiv.org/abs/2001.06782. Orthogonalization of gradients in general is an important technique and I am glad to see authors exploring down this direction.

However, the authors' claims do not hold up to serious vetting.

(1) The CIFAR baselines are incredibly weak. Could authors explain why?

(2) When SOTA CIFAR numbers are used, the proposed method falls short of SOTA numbers. Authors state "Unfortunately, the authors do not have the compute-power to extensively hyper-parameter tune a residual network" but then proceed to claim that it is "exceedingly likely" that such experiments would work out. Unfortunately, as reviewers we cannot just accept the authors' optimism that their method would hold up in those experiments without actual hard evidence.

(3) The general strategy of separating out the last dimension of gradients seems arbitrary. Why are those components important? Why not slice along another dimension? In other orthogonalizing gradient approaches, the full gradient tensor is considered, which assumes much less. Authors will need to more carefully justify this crucial design decision for their approach to pass muster.

(4) No related work section and a very sparse reference list. As mentioned, gradient orthogonalization is not necessarily novel, and should authors have done a more thorough literature search they may have become aware of this and been able to refine their method.

**Summary Of The Paper:**

Authors propose orthogonalizing gradient components across the last dimension of the gradient tensor. Authors show experiments for CIFAR10 and ImageNet and claim training speedup along with performance boosts when gradients are orthogonalized.

**Summary Of The Review:**

Major issues with clarity, algorithm design, and general structure of the writeup make it difficult to endorse this work.

---

> ### Author Response · Authors · 2021-11-22
> **Response to review**
>
> Thank you for taking the time to review our paper.
>
> (1) The CIFAR-10 baselines are from using SGD with "default" hyper-parameters, while we concede there is no set of parameters that work for every problem we have chosen a reasonable set that many practitioners would start off at.
> The high-performing hyper-parameter sets are gained through a thorough search to show-case the model's capacity, and are not wholly representative of how that optimisation would work applied to another task.
>
> (2) We agree that while we are optimistic there is no proof yet, however, hyper-parameter tuning cannot produce a set that give worse results than already shown, and that is not far off the best results.
> Additionally, see our new section on a comparison to Adam.
>
> (3) When deciding which dimension to orthogonalise over we have four choices, (channels_out, channels_in, conv_width, conv_height) of these only the channels_out is shared by the output representation (channels_out, image_width, image_height) therefore orthogonalising any other dimension would not help produce an orthogonalised representation.
> Our method lines up with previous literature e.g. [1].
>
> (4) While gradient orthogonalisation has been used in a multi-task setting, and weight orthogonalisation in a single-task setting, gradient orthogonalisation in a single-task setting is novel.
>
> [1] Huang, L., Liu, L., Zhu, F., Wan, D., Yuan, Z., Li, B. and Shao, L., 2020. Controllable orthogonalization in training dnns. In Proceedings of the IEEE/CVF Conference on Computer Vision and Pattern Recognition (pp. 6429-6438).

---

> > ### Comment · Reviewer_snPa · 2021-11-30
> > **RE: Response**
> >
> > Thanks to the authors for their response. I have read the response and reviewed the paper and have decided to keep my rating the same.

---

### Decision · Program_Chairs · 2022-01-20

**Decision:**

Reject

**Comment:**

This paper proposes orthogonalising loss gradients with respect to neural network parameters to speed up optimization and improve performance.

The reviewers are unanimous in recommending rejection of the paper. They highlight the following issues:
* weak baselines, which make it difficult to judge the contribution of this paper empirically
* lack of discussion of relevant literature and existing techniques
* arbitrary choices in the design of the algorithm, not backed up by theory or convincing arguments

The reviewers acknowledge the author response, but remain largely unconvinced of the merit of the proposed approach. I see no special reasons to disregard the reviewer assessments, and I therefore recommend not accepting this paper.